# Screening and treatment practices for iron deficiency in anaemic pregnant women: A cross-sectional survey of healthcare workers in Nigeria

Ochuwa Adiketu Babah [1,2,3,4] *, Lenka Beňová[4], Claudia Hanson[1], Ajibola Ibraheem Abioye[5], Elin C. Larsson[1], Bosede Bukola Afolabi[2,3,6]

1 Department of Global Public Health, Karolinska Institutet, Stockholm, Sweden, 2 Faculty of Clinical Sciences, College of Medicine, University of Lagos, Idi-Araba, Lagos, Nigeria, 3 Department of Obstetrics and Gynaecology, Lagos University Teaching Hospital, Idi-Araba, Lagos, Nigeria, 4 Department of Public Health, Institute of Tropical Medicine, Antwerp, Belgium, 5 Department of Global Health and Population, Harvard T.H. Chan School of Public Health, Boston, Massachusetts, United States of America, 6 Centre for Clinical Trials and Implementation Science (CCTRIS), College of Medicine, University of Lagos, Idi-Araba, Lagos, Nigeria

* ochuwa.babah@ki.se, obabah@unilag.edu.ng

**Data Availability Statement:** The dataset for this study is available in the open science framework

## Abstract

### Background

Iron deficiency anaemia in pregnancy is a significant contributor to maternal and perinatal morbidity and mortality globally. Despite international and national guidelines for its screening and treatment, knowledge and prescription practices of healthcare providers vary.

### Aim

To determine maternal healthcare workers' screening and treatment practices for iron deficiency in anaemic pregnancy women in two states in Nigeria.

### Methodology

This cross-sectional study sampled maternal healthcare workers from 84 randomly selected public health facilities in Lagos and Kano States. Data on methods of diagnosis and prescription practices for iron deficiency anaemia were collected using a self-administered questionnaire. Means and percentages were reported using probability weights, and a comparison of practices of anaemia treatment between doctors and nurses/midwives was done using Chi-square test or Fishers exact.

### Results

Of the 467 maternal healthcare workers surveyed (232 from Lagos, 235 from Kano), 40.0% were doctors, 54.0% nurses or midwives and 6.0% community health extension workers. In the sample, 27.6% always and 58.7% sometimes screened anaemic pregnant women for iron deficiency; among these, 84.7% screened using complete blood count. Oral iron for

(OSF) repository. The link to the dataset is https://osf.io/ybcdq/.

**Funding:** This work was supported, in whole, by the Bill & Melinda Gates Foundation [Investment ID INV-017271] grant for a clinical trial, IVON trial which principal investigator is Professor Bosede Afolabi, one of the PhD supervisors of the lead/corresponding author of the manuscript. The research was conducted as part of her doctoral training. The funding agency played no role in the conceptualization of this research idea, data collection, data analysis, or manuscript preparation.

**Competing interests:** The authors have declared that no competing interests exist.

**Abbreviations:** 95%CI, 95% confidence interval; CHEWs, Community health extension workers; IDA, Iron deficiency anaemia; IVON, Intravenous versus oral iron for iron deficiency anaemia treatment in pregnant Nigerian women; LGA, Local government area; MHW, Maternal healthcare worker; NDHS, National Demographic Health Survey; PHC, Primary Health Centre; REDCap, Research electronic data capture; WHO, World Health Organization.

treatment of iron deficiency anaemia was prescribed by 96.9%. Intravenous iron for treatment was prescribed by 30.2%, but by only by 18.6% as first-line drug (as iron dextran by 69.3% and as iron sucrose by 31.5% of intravenous iron prescribers). Commonest reasons for low usage of intravenous iron were cost and need for venepuncture. Fifty-three percent of maternal healthcare workers' prescribed iron supplements for anaemia during concomitant infection, with the prescription practice similar among doctors versus nurse/midwives (p = 0.074).

## Conclusion

We found suboptimal levels of screening for iron deficiency among anaemic pregnant women. Iron deficiency anaemia in pregnancy is almost exclusively treated with oral iron by maternal healthcare workers in the two Nigerian states, similarly between doctors and nurses/midwives. Further research into potential reasons for low screening for iron deficiency and low use of intravenous iron are needed.

## Introduction

Globally, anaemia complicates roughly one-third of pregnancies, which can lead to adverse outcomes for affected women and their babies [1]. The global prevalence of anaemia during pregnancy declined slightly from 41% in 2000 to 36% in 2019, suggestive of slow progress toward achieving the 2012 World Health Assembly target to halve anaemia in women of reproductive age by 2025 [2, 3]. Iron deficiency is the most common cause of anaemia in pregnancy. Worldwide, 52% of all pregnant women (anaemic and non-anaemic) are estimated to be iron deficient [4]. The burden of anaemia in pregnancy is higher in low- and middle-income countries (LMIC) compared to high-income countries (HIC), especially in sub-Saharan Africa and Southeast Asia, where 2 out of 5 women are likely to be anaemic during pregnancy [5].

The reason for high prevalence of iron deficiency anaemia (IDA) despite routine iron supplementation during pregnancy, especially in LMICs, is unclear. While oral iron is known to be effective for treatment of IDA, a sizeable proportion of iron-deficient pregnant women remain in a state of anaemia despite using oral iron supplements [6]. In addition, adherence to oral iron during pregnancy is suboptimal [7]. Earlier studies have linked suboptimal adherence to oral iron intake to gastrointestinal side effects; intake of high doses of oral iron is associated with higher incidence of side effects [8, 9]. However, there is no evidence on the prescription practices of alternative formulations like intravenous and intramuscular iron in sub-Saharan Africa.

The World Health Organization (WHO) established guidelines for prophylaxis and treatment of IDA in pregnancy [10, 11]; but we do not know how well these guidelines are followed in clinical practice in LMICs. A study in Burkina Faso evaluated all prescriptions issued to pregnant women for various indications, and reported suboptimal prescription practices of maternal healthcare workers (MHWs)–including cadres such as doctors, medical officers, nurses and midwives—despite availability of guidelines; less than 60% of prescriptions were identified as being appropriate in terms of choice and dosage of drug [12]. Ensuring that guidelines are appropriately translated into clinical practice is important to achieve progress in reducing the high burden of anaemia and IDA in pregnancy. It is important to understand the

prescription practices of MHWs because most pregnant women in sub-Saharan Africa obtain oral iron prescriptions from MHWs [13].

In Nigeria, IDA affects 25% - 46% of pregnant women (5). A recent study found a prevalence of 41% for iron deficiency among pregnant women with moderate or severe anaemia (haemoglobin concentration below 10g/dL) in the late second and early third trimesters of pregnancy in two states in Nigeria [14]. To the best of our knowledge, there were no guidelines on treatment of IDA in Nigeria at the time of this study [15]. In view of the fact that intravenous ferric carboxymaltose was recently tested for safety and efficacy in a clinical trial in Nigeria [16], evidence on screening and treatment practices of MHWs is critical to inform future clinical practice guidelines [6, 17–24]. The objective of this study is to determine the practice of key MHWs–doctors and nurses/midwives–in Nigeria regarding screening and treatment of iron deficiency in pregnant women with anaemia.

## Methodology

### Study setting

This is a cross-sectional study analysing a survey of MWHs in two states in Nigeria. The survey was conducted between 28[th] October 2022 and 6[th] January 2023 in Lagos state located in Southern Nigeria, and in Kano state located in Northern Nigeria. These are the two most populous states in Nigeria and differ substantially in the provision and utilisation of antenatal care (ANC) and childbirth services [25]. About 67% of pregnant women within the reproductive-age group receive ANC at least once from a doctor (17%) or a midwife (48%) according to the Nigeria Demographic and Health Survey 2018 [26].

### Participants

ANC is provided by a diverse group of healthcare workers in Nigeria. In some regions, especially in Northern Nigeria and rural areas in Southern Nigeria, midwives provide ANC and manage uncomplicated pregnancies and deliveries. The study sample included MHWs providing ANC in public health facilities in the two states. They included all cadres of doctors (medical officers, junior registrars, senior registrars, consultants—fully qualified specialists); nurses and midwives; and community health extension workers (CHEWs) who were permitted to prescribe medications at their health facility. A decision was made to include CHEWs in the survey when, at the commencement of fieldwork, we found that many primary health centres (PHCs), especially in Kano state, relied on CHEWs for the provision of ANC services because of a lack of doctors, nurses, and midwives.

### Sample size

Considering the proportion of doctors who prescribed oral iron as first-line drug for treatment of iron deficiency anaemia in pregnancy to be 77.7% (here assuming the same for midwives/nurses) as found in a previous study, [27] and using Cochrane's formula, [28] sample size was calculated as follows:

N = $Z^2pq/e^2$

Where: N = required sample size.

Z = abscissa of the normal curve that cuts off an area α at the tails, that is, the desired confidence interval 95% corresponding to a standard normal deviate of 1.96.

p = estimated proportion of an attribute, which was proportion of health workers who prescribed oral iron as first-line treatment of iron deficiency anaemia treatment in pregnancy = 77.7% (0.777) from a previous study [27]

q = 1 −p = 1–0.777 = 0.223

e = desired level of precision or error, 0.05

N = $1.96^2*0.777*0.223/0.05^2$ = 267

To adjust for design effect, we assumed a 50% difference in pre-service training or education between Northern and Southern medical and nursing/midwifery schools because we did not have existing data comparing pre-service training in these training institutions in Nigeria. The effective sample size was thus calculated by adding the assumed 50% (0.5) difference which made the design effect 1.0 + 0.5 = 1.5. So, we used a design effect of 1.5 to adjust the sample size,

Effective sample size = N*Deff

Where: N = required sample size calculated using Cochrane's formula = 267

Deff = Design effect = 1.5

Therefore, Effective sample size = 267*1.5 = 401

Adjustment was then made for 30% attrition, making sample size required for this study 522.

## Sampling

Nigeria has an estimated 24,640 doctors, 190,927 registered general nurses, 126,863 registered midwives and 549 licenced community midwives [29, 30]. The information on the number of MHWs by cadre employed in each health facility was not available before the survey. The size of the study population was approximated by the number of PHCs in each local government area (LGA), which assumed that on average a PHC employs the same number of MHWs and that the number of PHCs per LGA was directly related to the number of secondary and tertiary hospitals per LGA. The list of LGAs and public health facilities in Kano and Lagos states was obtained online from the Nigeria Health Facility Registry of the Federal Ministry of Health on 14th May 2022 [31]. All these facilities were listed as being publicly owned and providing maternal healthcare services at that time. The health facilities sampled comprised public health facilities and were categorised from lowest to highest as primary, secondary, and tertiary, according to the level of care.

A three-stage sampling technique was used in each state. In the first stage, we systematically selected ten LGAs with probability proportional to size (i.e., number of PHCs per LGA). This was done by arranging the LGA alphabetically, selecting a random number and calculating a sampling interval by dividing the total number of PHCs by 10. In the second stage, three PHCs and one secondary health facility were selected by simple random sampling technique from each of the ten selected LGAs per state. All tertiary institutions in both states were included in the sample. This meant that in both states, we sampled MHWs from 30 PHCs, ten secondary facilities, and all tertiary facilities (four in Lagos, three in Kano), with replacement if they were not operational, did not provide maternal care, or did not agree to grant access for this research. A list of replacement facilities was prepared during the initial sampling exercise. *Fig 1* shows the sampling frame.

In the third stage, to create the sampling frame of MHWs in each sampled health facility, the complete list of employed MHWs (all targeted cadres–doctors, nurses, midwives, and CHEWs) was obtained from the facility's administrative office or the MHWs duty roster if the former was unavailable. Potential respondents were selected using a simple random sampling technique (random number generation in Microsoft Excel). We selected and invited four respondents in each PHC, ten respondents in each secondary facility, and 16 respondents in each tertiary facility. This meant an anticipated total sample of 268 healthcare workers in Kano (120 in PHCs, 100 in secondary facilities, and 48 working in tertiary facilities) and 284 in

NIGERIA

| KANO STATE | | | | |
|---|---|---|---|---|
| Population | LGA | Level of care of public health facility | | |
| | | Primary | Secondary | Tertiary |
| Within state | 44 | 1216 | 48 | 4 |
| Study sample | 10 | 30 | 10 | 3 |

| LAGOS STATE | | | | |
|---|---|---|---|---|
| Population | LGA | Level of care of public health facility | | |
| | | Primary | Secondary | Tertiary |
| Within state | 20 | 412 | 44 | 4 |
| Study sample | 9 | 30 | 9 | 4 |

*Map created by the authors on Microsoft Excel; Powered by Bing ©Microsoft, OpenStreetMap. Created: 6 June 2024.*

**Fig 1. The sampling frame for the first and second stages of the multistage sampling.** LGA–Local government area. The target was to have 10 LGAs per state, but one LGA was selected twice in Lagos using systematic random sampling.

Lagos state (120 in PHCs, 100 in secondary facilities, and 64 in tertiary facilities); for a total combined sample of 552. The random selection of study participants was done centrally by the first author (OAB). MHWs who did not consent to participate in the survey were not replaced.

### Data collection

The questionnaire was adapted from a survey on the prescription practice of intravenous iron during pregnancy among Fellows of the Royal Australian and New Zealand College of Obstetricians (FRANZCOG) in Australia [27]. It had four sections: (A) sociodemographic characteristics of the respondent, including the years of professional experience and cadre of MHWs; (B) screening methods used for IDA in pregnancy used by respondents and the screening and diagnostic methods available at their health facilities; (C) questions about respondent's prescription practices for treatment of IDA in pregnancy; and (D) questions on how concomitant infections alter the respondent's practice of anaemia treatment in pregnancy.

The questionnaire was pretested on 13 MHWs with diverse backgrounds in Lagos State and indicated good face validity and a Cronbach's alpha scale reliability coefficient of 0.751. Minor modifications were made where necessary, based on the findings, comprehension of the questions, length, and any other feedback during pre-testing. The estimated time to fill in the questionnaire was 5–10 minutes for 75% of the pretest survey participants. The responses obtained during pre-testing were excluded from the final analysis sample. The final questionnaire is *S1 File*.

One research assistant was employed in each state to administer the survey under the supervision of a research doctor (one doctor employed for Kano state and OAB for Lagos). All members of the research team were trained by OAB. The survey was self-administered using an electronic data collection tool (tablet) with a Research Electronic Data Capture (REDCap) survey link. It was completed by consenting participants during working hours in a private space at their place of work. In addition to the electronic tablet-based modality, paper copies of the questionnaire were made available in case of challenges such as lack of internet connectivity or for eligible healthcare workers too busy to input data on the electronic platform directly at the time of the survey. All data collected from the survey were downloaded onto a password-protected database. All data obtained on paper copies were entered into the same database by the doctor overseeing activities in each state.

## Outcome measures

The primary outcome measure was the proportion of MHWs who prescribe oral, intravenous, and intramuscular iron preparations for the treatment of IDA in pregnancy. Secondary outcome measures were the proportion of MHWs who prescribe iron preparations when there is a concomitant infection and the proportion of MHWs who report always screening anaemic pregnant women further for iron deficiency before treatment.

### Definitions.

- Proportion of MHWs who prescribe oral, intravenous, or intramuscular iron preparations for the treatment of IDA in pregnancy refers to the number of maternal healthcare workers who prescribe oral, intravenous, or intramuscular iron as first line medication for treatment of IDA in pregnancy divided by total number of maternal healthcare workers surveyed (or total number of respondents) multiplied by 100 and expressed as a percentage.

- Proportion of maternal healthcare workers who give iron preparations when there is an ongoing infection refers to the number of maternal healthcare workers who would give iron preparations to their patients when there is concomitant infection divided by the total number of maternal healthcare workers surveyed or respondents multiplied by 100 and expressed as a percentage.

- Proportion of maternal healthcare workers who screen anaemic pregnant women further for iron deficiency before treatment refers to the number of maternal healthcare workers who screen pregnant women with anaemia for iron deficiency before commencing treatment divided by the total number of maternal healthcare workers surveyed who screen pregnant women for anaemia multiplied by 100 and expressed as a percentage.

## Data analysis

Data were analysed using STATA version 16.1 (Stata Corp. 2019. Stata Statistical Software: Release 16. College Station, TX: Stata Corp LLC.). Sixty-seven duplicates were removed during data cleaning using unique identifiers like eligibility screening, consent, age, gender, state, facility type, cadre of MHW, duration of medical practice, and duration of maternal care provision. Most duplicates were due to internet failure during completion of the survey by participants. Descriptive statistical analysis was done and continuous variables such as MHWs' age and years of experience were presented as mean ± standard deviation (SD). Categorical variables such as cadre and rank of MHWs and level of healthcare facility where the respondent primarily practiced were presented as frequencies and percentages. The primary outcome measure was presented as a percentage with a 95% confidence interval (CI).

We compared the practices of the two main cadres of MHWs–doctors versus nurses/midwives (CHEWs are excluded from such comparisons) in terms of screening and prescription practices for IDA in pregnancy and differences in MHWs characteristics between the two states using chi-square test, or Fishers exact test where the expected value in more than 25% of the cells in the contingency table is less than 5. Missing values were excluded from the analysis of each variable. There were ten (2.1%) values missing for age, one (0.2%) for gender, one (0.2%) for facility type, seven (1.5%) for duration of practice, four (0.9%) for duration of providing maternal care, and 28 (6.0%) for screening practice. Statistical significance was set at $p < 0.05$ using a two-tailed hypothesis.

In all analyses, we adjusted for clustering at the level of LGA and for stratification based on level of health facility. This was a self-weighting sample of health facilities, but because of the

selection of different numbers of respondents per facility type, we adjusted for unequal probability of being selected by applying weights during analysis. We generated inverse probability weights for respondents based on the estimated number of states within the country (p1), the number of health facilities per state (p2) and the number of MHWs in each level of health facility (p3). The weights were calculated as the inverse of the sum of the probabilities obtained, 1/(p1+p2+p3); and the participants' individual weight was calculated by dividing the weight for each individual by the mean weight [32, 33].

### Ethical consideration

Ethical approval was obtained from Lagos University Teaching Hospital HREC (Approval No. ADM/DSCST/HREC/APP/4864); Lagos State University Teaching Hospital HREC (Approval No. LREC/06/10/1916), Ikeja, Lagos; Federal Medical Centre, Ebute-Metta, Lagos (Approval No. HREC 22–11); and Aminu Kano Teaching Hospital HREC, Kano (Approval No. NHREC/28/01/2020/AKTH/EC/3405) before commencing this survey. Permission was also sought from Lagos State Health Service Commission (Approval No. LSHSC/2222/VOLIV/69); Primary Healthcare Board Lagos (Approval No. LS/PHCB/DPRS/256/VOL.I/064); and State Ministry of Health, Kano (Approval No. NHREC/17/03/2018) for use of the primary and secondary health facilities within the states.

Informed consent was obtained from the MHW before being allowed to participate in the survey. The research was performed in accordance with the principles of human research ethics as in the Declaration of Helsinki. Confidentiality was maintained in the survey. Identifiers like individual or institutional names and email addresses of participants were not requested. Participation was completely voluntary. All participants consented orally first following counselling based on the study information in the informed consent form (*S2 File*), and thereafter on REDCap before they could proceed with the survey. Those who completed paper questionnaire consented by signing a written document.

### Results

We invited 488 participants. The reasons for the lower number compared to the targeted sample size (n = 522) was because fewer than expected MHWs worked at some health facilities; non-functionality of one health facility with no possible replacement within the selected LGA; and non-provision of maternal care at one tertiary health facility in Lagos. Of the 488 invited MHWs, ten participants declined and a further 11 did not complete the survey, resulting in a total of 467 valid responses: 232 from Lagos (97.9% response rate), and 235 from Kano (93.6% response rate). *Fig 2* shows the flow chart for inclusion of the MHWs in the survey. Most (75.4%) responded electronically on REDCap and 115 (24.6%) had data collected using paper copy.

As shown in *Table 1*, half of the participants were registered nurses/midwives (54.1%), 39.8% doctors, and 6.1% CHEWs. The mean age of respondents was 36.1 years, 27.8% were male and 72.2% female. The mean duration of provision of maternal healthcare by the MHWs was 8.3 years.

We found that among doctors and nurses/midwives (n = 370), 27.6% (95%CI: 23.2–32.6%) always screening pregnant women with anaemia for iron deficiency (*Table 2*), with no significant difference between the two cadres (p = 0.469). Compared to nurses/midwives, a significantly higher percentage of doctors reported screening for IDA using complete blood count, peripheral blood film, and serum ferritin; whereas nurses/midwives were more likely to screen using iron profile.

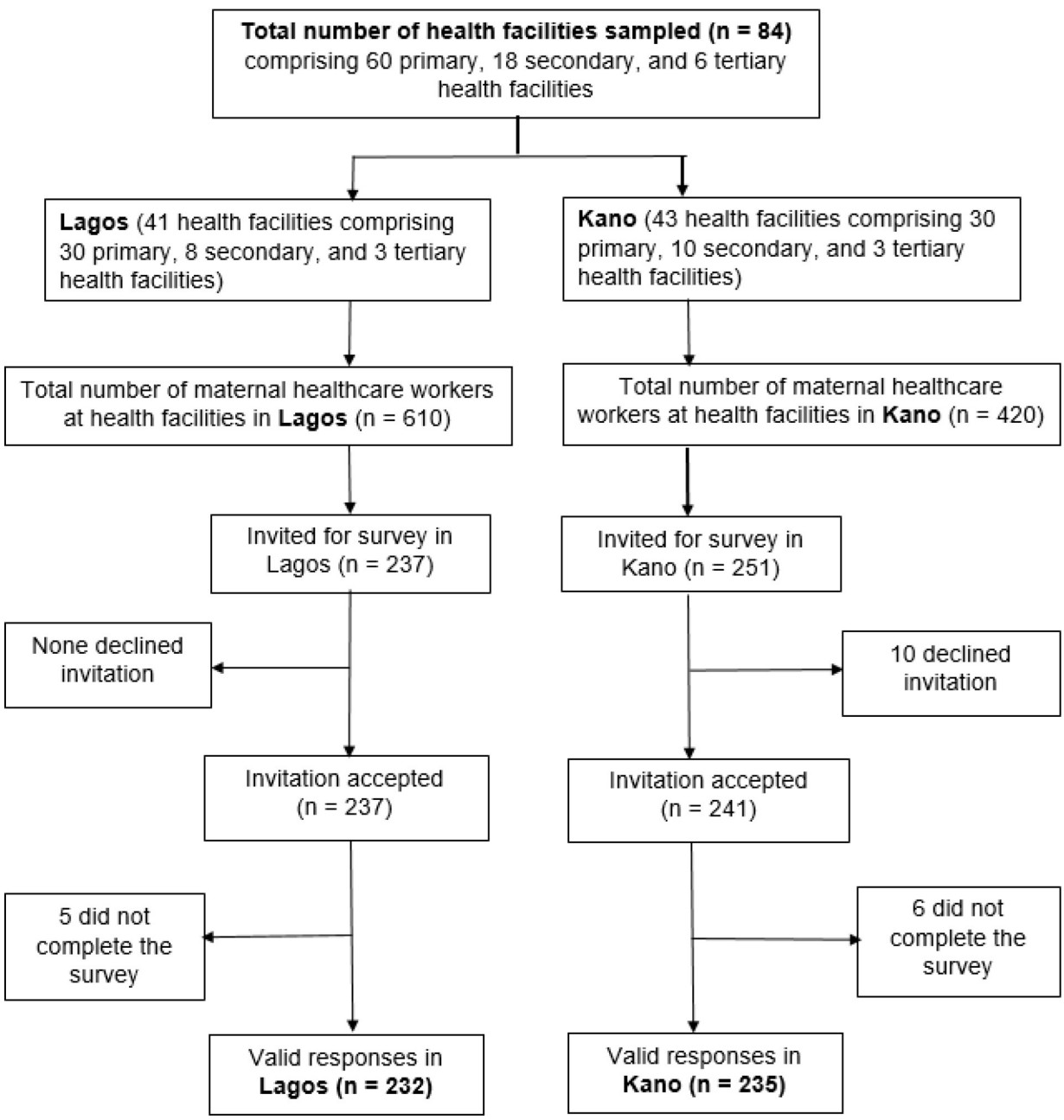

**Fig 2. Flow diagram for respondents' inclusion in the survey.** Note: In Lagos state the health facilities were two fewer than earlier planned because one secondary health facility was non-functional with no replacement available and one tertiary health facility was not providing maternal care.

Among all cadres of respondents, those who reported 'not always' or 'never' screening anaemic pregnant women for IDA (n = 316), the reasons for not screening included the presumption that iron deficiency is the most probable cause of anaemia (44.2%), delay in getting test results (31.3%), cost of screening test (29.9%), and lack of laboratory facility (29.6%) as shown in *Fig 3*. More nurses/midwives (81.1%) compared to doctors (47.4%) reported lack of laboratory facilities for not screening for iron deficiency in anaemic pregnant women.

**Table 1. Sociodemographic profile of sample of maternal healthcare workers in Lagos and Kano States, Nigeria (n = 467).**

| Sociodemographic characteristic | All participants<br>n = 467 | Lagos State<br>n = 232 | Kano State<br>n = 235 |
|---|---|---|---|
| | Mean (95%CI) | Mean (95%CI) | Mean (95%CI) |
| **Age in years,** n = 457 | 36.1 (35.1–37.1) | 38.3 (37.0–39.7) | 33.1 (31.8–34.5) |
| **Duration of practice post qualification in years,** n = 460 | 9.9 (9.0–10.7) | 11.3 (10.1–12.5) | 8.0 (6.8–9.2) |
| **Duration of provision of maternal care in years,** n = 463 | 8.3 (7.5–9.0) | 9.5 (8.4–10.7) | 6.6 (5.6–7.6) |
| | % (95% CI) | % (95% CI) | % (95% CI) |
| **Cadre** | | | |
| Doctor (n = 132) | 39.8 (35.4–44.5) | 45.8 (39.3–52.4) | 32.6 (26.6–39.0) |
| Registered nurse/midwife (n = 238) | 54.1 (49.5–58.7) | 49.1 (42.5–55.7) | 60.2 (53.7–66.6) |
| Community health extension worker (n = 97) | 6.1 (4.1–8.6) | 5.1 (2.7–8.8) | 7.2 (4.2–11.3) |
| **Gender** | | | |
| Male (n = 135) | 27.8 (23.8–32.1) | 27.7 (22.0–33.9) | 27.9 (22.3–34.1) |
| Female (n = 331) | 72.2 (67.9–76.2) | 72.3 (66.1–78.0) | 72.1 (65.9–77.7) |
| **Number of anaemic pregnant women seen weekly** (n = 464) | | | |
| 0–5 | 67.2 (62.7–71.5) | 83.3 (77.8–87.8) | 47.3 (40.7–54.0) |
| >5 | 32.8 (28.5–37.3) | 16.7 (12.2–22.2) | 52.7 (46.0–59.3) |

**Table 2. Self-reported practices regarding screening and diagnosis of IDA in pregnancy among doctors versus nurses/midwives in Lagos and Kano states in Nigeria.**

| Variable | Total number | % Total<br>n = 370 | % Doctors<br>n = 132 | % Registered nurses and/or midwives<br>n = 238 | p-value |
|---|---|---|---|---|---|
| **Screen for iron deficiency in pregnant women with anaemia (n = 356)** | | | | | |
| Always | 96 | 27.6 (23.0–32.6) | 22.5 (15.6–30.7) | 31.4 (25.4–37.9) | 0.469^ |
| Sometimes | 210 | 58.7 (53.4–63.8) | 63.0 (54.1–71.4) | 55.4 (48.7–62.0) | |
| Never | 50 | 13.7 (10.3–17.7) | 14.4 (8.9–21.7) | 13.2 (9.1–18.3) | |
| **Screening methods for IDA among those who screen (multiple responses allowed)**\*\* | | | | | |
| *Complete blood count (n = 300)* | | | | | |
| Yes | 253 | 84.7 (80.1–88.6) | 96.9 (91.6–99.3) | 75.5 (68.7–81.5) | <0.001^ |
| No | 47 | 15.3 (11.4–19.9) | 3.1 (0.7–8.4) | 24.5 (18.5–31.3) | |
| *Peripheral blood film (n = 295)* | | | | | |
| Yes | 87 | 37.5 (31.9–43.3) | 56.3 (46.4–65.9) | 23.2 (17.3–29.9) | <0.001^ |
| No | 208 | 62.5 (56.7–68.1) | 43.7 (34.1–53.6) | 76.8 (70.1–82.7) | |
| *Serum ferritin (n = 294)* | | | | | |
| Yes | 77 | 29.2 (24.1–34.8) | 36.0 (27.0–45.9) | 24.1 (18.1–30.9) | <0.001^ |
| No | 217 | 70.8 (65.2–75.9) | 64.0 (54.1–73.0) | 75.9 (69.1–81.9) | |
| *Iron profile (n = 296)* | | | | | |
| Yes | 90 | 33.7 (28.3–39.4) | 13.6 (7.8–21.6) | 48.9 (77.5–94.3) | <0.001^ |
| No | 206 | 66.3 (60.6–71.7) | 86.4 (78.4–92.2) | 51.1 (43.7–58.5) | |

Variables are presented as percentages (95%CI) of the weighted sample. Total comprises doctors and nurses/midwives. Fourteen non-responses on screening for iron deficiency. p-values compare doctors versus nurses/midwives only. IDA—iron deficiency anaemia. %—percentage of. ^Chi square test. #Fishers exact test.

\*\*Of the unweighted sample of 306 doctors and nurses/midwives who screen for IDA always or sometimes, 124 (40.5%) screen with only one laboratory method, 100 (32.7%) screen with either of two methods, 37 (12.1%) screen with either of three methods, 18 (5.9%) screen using any of the four methods listed, while 27 (8.8%) did not select any method.

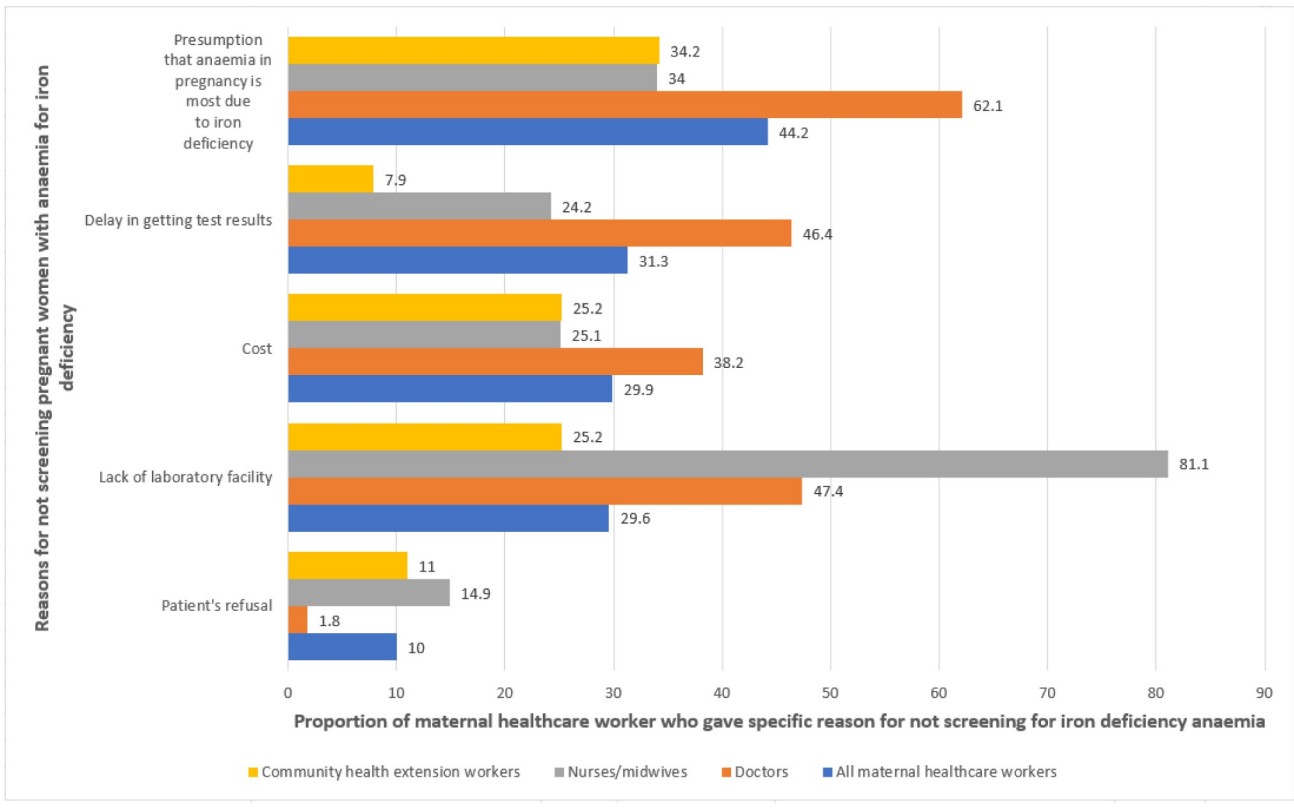

**Fig 3. Reasons for not screening for iron deficiency anaemia in pregnancy, among respondents who not always/never screen pregnant women with anaemia for iron deficiency (n = 316).** Twenty-eight healthcare workers who did not provide any answers were excluded from the analysis. Multiple responses were allowed. Of the unweighted sample of 316 respondents who gave reasons for not screening, 173 (54.7%) gave only one reason, 87 (27.5%) two reasons, 43 (13.6%) gave three reasons and 9 (2.8%) gave more than three reasons, while 4 (1.3%) selected "No" for all reasons and did not give any other reason.

*Table 3* shows the prescription practices in terms of the types of iron formulation used for treating IDA in pregnancy and the prescription order. Among the doctors and nurses/mid-wives, 96.9% reported that they usually prescribe oral iron for pregnant women with IDA. The use of intravenous or intramuscular iron was uncommon, with 30.2% of doctors and nurses/midwives reporting ever prescribing intravenous iron and 10.3% intramuscular iron for treatment of IDA, with no significant difference between cadres. Compared to doctors, a significantly higher percentage of nurses/midwives reported that they do not prescribe any treatment (0.0% vs. 4.0%, p = 0.016) or referred anaemic pregnant women to other facilities for treatment (1.3% vs. 18.8%, p = 0.0104). In terms of prescription order, oral iron was most commonly prescribed as a first line drug for IDA treatment by 86.3% of doctors and nurses/midwives. Among doctors and nurses/midwives who prescribed oral or intramuscular iron as a first line drug, most of them (18.9% and 26.7%, respectively) prescribed intravenous iron as second line drug. Among those who prescribed intravenous iron as first line drug, most (41.4%) prescribed oral iron as second line drug.

Further analysis of doctors and nurses/midwives showed that the most common forms of oral iron (tablet or capsule) prescribed by respondents were ferrous sulphate 35.8% (31.4–40.3%), ferrous fumarate 21.8% (18.1–25.8%), and ferrous gluconate 20.6% (17.0–24.5%). Of those who prescribed ferrous sulphate, 57.3% reported prescribing dosages above WHO recommended levels of 120mg elemental iron daily for treatment of IDA during pregnancy.

**Table 3. Prescription practices for treatment of iron deficiency anaemia in pregnancy of doctors and nurses/midwives in Nigeria (n = 370).**

| Variable | % Total<br>% (95%CI)<br>n = 370 | Doctors<br>% (95%CI)<br>n = 132 | Nurses and/or midwives<br>% (95%CI)<br>n = 238 | p-value |
|---|---|---|---|---|
| **Type of iron preparation prescribed (multiple response options)** | | | | |
| Oral iron | 96.9 (93.9–98.7) | 99.1 (94.4–100.0) | 95.3 (90.7–98.0) | 0.162[#] |
| Intravenous iron | 30.2 (24.5–36.3) | 33.5 (23.8–44.3) | 27.7 (20.9–35.3) | 0.211^ |
| Intramuscular iron | 10.3 (6.8–14.7) | 11.6 (5.8–20.2) | 9.3 (5.2–14.8) | 0.162^ |
| Refer to another healthcare provider | 11.4 (7.8–16.0) | 1.3 (0.1–6.4) | 18.8 (13.1–25.7) | **0.004^** |
| Do not prescribe any treatment | 2.3 (0.8–5.0) | 0.0 | 4.0 (1.5–8.3) | **0.016[#]** |
| **Prescription order of iron preparations** (n = 370) | | | | |
| *First line: Oral iron* | 86.3 (82.0–89.9) | 82.3 (74.2–88.7) | 89.5 (84.4–93.4) | 0.089^ |
| *Second line (n = 277)* | | | | |
| Intravenous only | 18.9 (14.5–24.0) | 20.5 (13.1–29.8) | 17.7 (12.4–24.1) | 0.457[#] |
| Intramuscular only | 2.1 (0.7–4.5) | 2.3 (0.3–7.6) | 1.9 (0.4–5.1) | |
| Intravenous and intramuscular | 5.0 (2.7–8.2) | 5.4 (1.9–11.9) | 4.6 (2.0–8.8) | |
| No option selected | 74.1 (68.5–79.2) | 71.7 (61.8–80.3) | 75.8 (68.8–81.9) | |
| *First line: Intravenous iron* | 18.6 (10.8–28.8) | 26.9 (13.0–45.1) | 11.9 (4.3–24.5) | 0.400^ |
| *Second line (n = 16)* | | | | |
| Oral iron only | 41.4 (18.0–68.1) | 37.0 (8.3–75.1) | 49.4 (15.3–83.9) | 0.619[#] |
| Intramuscular only | 5.5 (0.1–29.1) | 0.0 | 15.4 (0.7–55.8) | |
| Oral and intramuscular | - | - | - | |
| No option selected | 53.2 (27.3–77.9) | 63. 0 (24.9–91.7) | 35.2 (7.4–73.7) | |
| *First line: Intramuscular iron* | 10.7 (2.7–26.9) | 15.7 (2.5–43.2) | 6.6 (0.2–29.7) | 0.645[#] |
| *Second line (n = 5)* | | | | |
| Oral iron only | - | - | - | >0.999[#] |
| Intravenous only | 26.7 (1.5–76.8) | 39.4 (1.7–92.9) | 0.0 | |
| Oral and intravenous | - | - | - | |
| No option selected | 73.3 (23.2–98.5) | 60.6 (7.1–98.3) | 100.0 | |

Total comprises all doctors, nurses, and midwives only. Sample included for analysis of second line drug prescription comprise only the doctors and nurses/midwives who prescribe the corresponding first line drug. ^Pearson chi-square test. [#]Fisher's exact test. p-value compares the prescription practices of doctors versus nurses/midwives.

Among prescribers of ferrous gluconate, 4.0% prescribed dosages above WHO recommendation. *S1 and S2 Tables* provide a detailed description on the pattern of prescription of oral iron. Among prescribers of intravenous iron, the commonest formulations used were iron dextran by 69.3% (43.6–88.5%) and iron sucrose by 31.5% (12.1–57.2%). *S1 Fig* shows prescription practices for oral, intravenous, and intramuscular iron, by trimester of pregnancy. Though all the MHWs reported prescribing at least an iron formulation at any trimester of pregnancy, 73.0% prescribed oral iron in the first trimester whereas less than 6.0% ever prescribed intravenous or intramuscular iron in first trimester. The percentage who reported ever prescribing intravenous or intramuscular iron increased in second and third trimesters of pregnancy. *S3 Table* shows the main advantages of the use of intravenous iron for IDA treatment from the MHWs' perspective and these were faster improvement in anaemia treatment compared to oral iron 47.3%, its suitability for faster correction of IDA in late pregnancy 34.5% and as second-line treatment when oral iron fails 31.2%. The main disadvantages were higher cost 40.1%, need for venepuncture 39.8% and the need for human resources to administer the drug 25.9%.

We looked at IDA treatment when there is a concomitant infection. We found that among all MHWs, 52.8% (95%CI: 47.6–57.9%) reported treating anaemia using iron supplements when there was a concomitant infection, with no significant differences between doctors and nurses/midwives (p = 0.074). In the presence of infection, the preferred treatment option was oral iron (by 42.1% of respondents), blood transfusion (38.4%), intramuscular iron (11.4%), intravenous iron (3.5%), and dietary modification (0.4%), S4 Table. For those who treated IDA with iron supplements when there was a concomitant infection, the main reasons for doing so were to avert complications 26.4% (95%CI: 18.9–34.9%) and because of fear of anaemia worsening 16.3% (95%CI: 10.3–24.0%). For those who do not give iron supplements when there is a concomitant infection, the reasons included that referral was indicated to higher centres or more experienced health workers 38.2% (95%CI: 30.3–46.8%) and that iron favours microbial growth 36.6% (95%CI: 28.7–45.0%), S2 Fig.

## Discussion

This study of screening and prescription practices for iron deficiency among anaemic pregnant women in Nigeria found that only a quarter of MHWs report screening anaemic pregnant women consistently for iron deficiency. The majority of MHWs prescribed oral iron for treatment of IDA in pregnancy. Prescriptions of intravenous and intramuscular iron as first line treatment were uncommon. There is a lack of consensus regarding the concurrent administration of iron supplements to anaemic pregnant women who have a concomitant infection. The practices of screening anaemic pregnant women for iron deficiency and drug prescription for IDA treatment during pregnancy were similar among doctors and nurses/midwives.

Despite the high prevalence of anaemia and IDA in pregnancy globally, especially in South-East Asia and sub-Saharan Africa [34], this study found a lower screening rate for IDA in pregnancy in Nigeria, a low-middle-income country, compared to a high-income country (United States of America) where as many as 50% of obstetricians and gynaecologists screen all pregnant women routinely for IDA [35]. The low screening rate in our study was attributed majorly to the presumption that most cases of anaemia are due to iron deficiency and did not require overcoming existing delays in obtaining test results and their high out-of-pocket cost. The latter emphasizes the need to explore the possibility of implementing the use of rapid and affordable diagnostic test kits for screening for iron deficiency in regions of high burden for IDA in pregnancy.

Regarding treatment of IDA in pregnancy, MHWs preference for oral iron is possibly due to the perception that intravenous iron preparations are expensive and require venepuncture and additional human resources despite the perceived benefit of faster anaemia correction and suitability as a second line agent. This is not surprising considering the inequity in the healthcare system in sub-Saharan Africa including Nigeria in terms of unavailability and inequitable distribution of healthcare workers [36]. It also supports the report by the MHWs on the need for human resources as a perceived disadvantage for intravenous iron administration. These barriers identify the need for deeper exploration into the barriers and facilitators of intravenous iron use.

In addition, most MHWs were less comfortable giving parenteral iron in the first compared to the third trimester of pregnancy. This finding is corroborated by an earlier study in Australia which evaluated the prescription practices of Obstetricians and found that only 8% of the Obstetricians would prescribe intravenous iron in the first trimester, even though 96% of them prescribe intravenous iron for treatment of pregnant women with IDA [27]. The reservation against using intravenous iron in the first trimester of pregnancy is likely because there is limited evidence regarding the teratogenic risk associated with the use of intravenous iron in the

first trimester of pregnancy [37]. The predominant use of iron dextran by those studied is surprising considering that studies have found a relatively high incidence of anaphylactic reactions with the use of high molecular weight iron dextran preparations [38]. This may be due to a lack of awareness of this safety concern especially as only 25% of respondents reported safety issues as a perceived disadvantage of IV iron use. It may also be because of unavailability of suitable intravenous or intramuscular alternatives in the health facilities or pharmacies.

Our finding that half of the MHWs treat anaemic pregnant women with iron supplements in the presence of infection suggests that there is no consensus on whether to prescribe iron when there is a concomitant infection. A study by Jonker et al in sub-Saharan Africa provided evidence that adverse outcomes increase in children with infections like malaria and diarrhoeal diseases when supplemented with oral iron, but there is no firm evidence of such a risk in pregnancy [39]. We found that some MHWs reported withholding iron supplements when there was concomitant infection because iron favours microbial growth which might worsen the infection [40]. This finding is corroborated by a recent systematic review which provided moderate evidence of an increased risk of infection with the use of intravenous iron [41]. However, the findings of this systematic review have been criticised because most of the studies included which evaluated the effect of intravenous iron on infection were effectiveness studies, and did not have infection as an endpoint [42]. Contrary to earlier reports, Mwangi et al in their review, found substantial benefits with iron supplementation when there is a concomitant infection in iron-deficient women [43]. There is a need for further studies to evaluate the impact of anaemia treatment on recovery from infection. It is also important to develop clear guidelines for MHWs to translate evidence-based findings into clinical practice.

The similarities in screening and treatment practices for IDA during pregnancy between doctors and nurses/midwives in Nigeria are most likely because nurses/midwives often consult doctors, especially regarding pathology and pharmacological interventions [44]. The finding that more nurses/midwives compared to doctors reported unavailability of laboratory facilities for screening for iron deficiency may suggest a lack of awareness by the nurses of available laboratory facilities because they are less empowered to offer care in the healthcare system. In addition, doctors have better exposure and knowledge regarding the treatment of medical conditions because their training curriculum is more elaborate than that of the nurses. We are not quite certain if attitude may play a role. This observation is relevant because of our finding that CHEWs provide ANC, suggesting that there is task-shifting to lower cadres within the healthcare system. This emphasizes the need for a high-quality guideline for more effective service delivery at all levels. Considering that MHWs tend to prescribe higher than recommended doses of oral iron as found in this study, suggesting a lack of awareness about the WHO guideline on anaemia in pregnancy or its usage, the need for training and re-training of MHWs is important.

## Strengths and limitations

This study involved MHWs from the two most populous of the thirty-six states in the country and used multistage sampling to achieve representativeness of the sample. The response rate in the survey was high because each MHW was physically approached, rather than using an online survey, which is associated with a high refusal and low completion rates. In addition, the survey did not contain sensitive, identifying information that may discourage the MHW from participating or giving honest responses. However, we cannot rule out response-choice order effects and recall bias which are risks associated with self-administered questionnaires [45]. To the best of our knowledge, this study would be the first in sub-Saharan Africa to evaluate the screening and treatment practices of MHWs for IDA during pregnancy.

## Conclusion

This study of MHWs in two large states in Nigeria showed that screening for iron deficiency in pregnant women with anaemia is low and that iron deficiency anaemia in pregnancy is almost exclusively treated with oral iron. The use of intravenous iron as a first line agent for IDA treatment in pregnancy is very low. Further research into the potential reasons for low screening for iron deficiency and low usage of intravenous iron are needed. In addition, determining the cost-effectiveness of intravenous iron and factors affecting its routine use for the treatment of IDA in pregnancy in Nigeria is required.

## Supporting information

**S1 File. Survey questionnaire.**
(DOCX)

**S2 File. Information sheet and consent form.**
(DOCX)

**S1 Table. Type and dosages of oral iron prescribed by maternal healthcare workers.**
(DOCX)

**S2 Table. Dosage of iron tablets commonly prescribed for the treatment of iron deficiency anaemia.**
(DOCX)

**S3 Table. Perception of maternal health workers regarding the use of intravenous iron for treatment in pregnancy (n = 467).**
(DOCX)

**S4 Table. Maternal health workers' approach towards treating iron deficiency anaemia when there is concomitant infection.**
(DOCX)

**S1 Fig. Maternal healthcare workers' willingness to prescribe various formulations of iron in each trimester of pregnancy.**
(DOCX)

**S2 Fig. Maternal health workers' perception towards treating iron deficiency anaemia when there is concomitant infection.**
(DOCX)

## Acknowledgments

Appreciations to Dr Abdulazeez and Dr Abdulrasheed for the joint effort in facilitating data collection for this study at the various health facilities in Kano State. Appreciations to Mrs Elizabeth Ekwote and Mrs Elizabeth Harrison for assisting with data collection at the health facilities in Lagos State.

## Author Contributions

**Conceptualization:** Ochuwa Adiketu Babah, Lenka Beňová, Claudia Hanson, Elin C. Larsson, Bosede Bukola Afolabi.

**Data curation:** Ochuwa Adiketu Babah.

**Formal analysis:** Ochuwa Adiketu Babah, Ajibola Ibraheem Abioye.

**Funding acquisition:** Bosede Bukola Afolabi.

**Investigation:** Ochuwa Adiketu Babah.

**Methodology:** Ochuwa Adiketu Babah, Lenka Beňová, Claudia Hanson, Ajibola Ibraheem Abioye, Elin C. Larsson, Bosede Bukola Afolabi.

**Project administration:** Ochuwa Adiketu Babah.

**Software:** Ajibola Ibraheem Abioye.

**Supervision:** Lenka Beňová, Claudia Hanson, Elin C. Larsson, Bosede Bukola Afolabi.

**Validation:** Ochuwa Adiketu Babah.

**Visualization:** Ajibola Ibraheem Abioye.

**Writing – original draft:** Ochuwa Adiketu Babah.

**Writing – review & editing:** Ochuwa Adiketu Babah, Lenka Beňová, Claudia Hanson, Ajibola Ibraheem Abioye, Elin C. Larsson, Bosede Bukola Afolabi.

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
