## [Decision Letter · Decision Letter 0]

2 Jun 2024

PONE-D-24-09943Screening and treatment practices for iron deficiency anaemia during pregnancy: a cross-sectional survey of healthcare workers in NigeriaPLOS ONE

Dear Dr. Babah,

Thank you for submitting your manuscript to PLOS ONE. After careful consideration, we feel that it has merit but does not fully meet PLOS ONE’s publication criteria as it currently stands. Therefore, we invite you to submit a revised version of the manuscript that addresses the points raised during the review process.

Please see the comments of two reviewers below. In addition, please include the full ethics statement in the Methods section, and please ensure that you have reported all covariates and confounding variables.

We look forward to receiving your revised manuscript.

Kind regards,

Hanna Landenmark

Staff Editor

PLOS ONE

2. In this instance it seems there may be acceptable restrictions in place that prevent the public sharing of your minimal data. However, in line with our goal of ensuring long-term data availability to all interested researchers, PLOS’ Data Policy states that authors cannot be the sole named individuals responsible for ensuring data access (http://journals.plos.org/plosone/s/data-availability#loc-acceptable-data-sharing-methods).

Reviewers' comments:

Reviewer's Responses to Questions

**Comments to the Author**

1. Is the manuscript technically sound, and do the data support the conclusions?

Reviewer #1: Yes

Reviewer #2: Yes

2. Has the statistical analysis been performed appropriately and rigorously? 

Reviewer #1: Yes

Reviewer #2: Yes

3. Have the authors made all data underlying the findings in their manuscript fully available?

Reviewer #1: Yes

Reviewer #2: Yes

4. Is the manuscript presented in an intelligible fashion and written in standard English?

Reviewer #1: Yes

Reviewer #2: Yes

5. Review Comments to the Author

Reviewer #1: Is it "Screening and treatment practices for IDA during pregnancy among ANC visitors or among anaemic pregnant mothers?" Clearly state and write it from the result and discussion sections depending on your aim.

You state various keywords even more than the recommended.

You used abbreviations/acronyms at the abstract section.

Show clearly how you calculated sample size.

Why do you consider an attrition rate of 30%? How much is the recommendation?

Put your operational definitions.

More nurses/midwives (81.1%) compared to doctors (47.4%) reported lack of laboratory facilities for not screening for iron deficiency in anaemic pregnant women. Compared to doctors, a significantly higher percentage of nurses/midwives reported that they do not prescribe any treatment. Do you consider any knowledge and attitude issue in addition to referral practice?

Why do you concern in low usage of intravenous iron than oral iron?

Reviewer #2: Thorough background research on the state of maternal health care workforce in Nigeria. Good job by the research team. More specific references for those values of the population of maternal healthcare professionals would be better.

6. PLOS authors have the option to publish the peer review history of their article (what does this mean?). If published, this will include your full peer review and any attached files.

Reviewer #1: No

Reviewer #2: **Yes: **Dr. Hameedat Opeyemi Abdussalam.

---

## [Author Response · Author response to Decision Letter 0]

17 Jun 2024

TO THE EDITOR

1. Thank you for your guidance. Ethics statements have been moved from the section on declaration to the section on ethical consideration in the methodology (Page 13, lines 287-299).

2. Manuscript has also been checked to ensure all covariates and confounding variables are reported.

3. A response letter to the reviewers’ comments, a marked copy of the manuscript (tracked), and a clean copy of the manuscript have been uploaded to portal.

4. We do not have plans to make any change to the financial disclosure.

5. We did not find a need to publish the protocol for this study since it is not laboratory-based research. However, to enhance reproducibility of the research, we have extensively described how the research was conducted in the Methods section, and also include the complete survey questionnaire and informed consent form in the appendix.

6. The journal style is well noted. We have gone through the entire manuscript to ensure that the style is in order. Revisions have been made where necessary. These include recommended symbol labels for authorship (Page 1, lines 4 and 5) and use of email only for corresponding author (Page 1, lines 15-20). In addition, we have ensured that all titles and sub-titles have the recommended fonts, and that tables, figures and supplementary information files, tables and figures are presented and labelled according to the journal requirements.

7. We have deposited the dataset for this study in Open Science Framework. The data is accessible at: https://osf.io/ybcdq/

8. Thank you for bringing forth an observation on the need to cite the source of the map in Figure 1. The previous image was created using an editable map obtained from a colleague. Unfortunately, we could not trace the source. So, we have now created a new map of Nigeria, showing the two states where the study was conducted, on Microsoft Excel. The facility for generating map on Excel spreadsheet is powered by Bing © Microsoft OpenStreetMap and has been cited in Figure 1, with the date the map was created indicated (Figure 1).

9. In addition to the newly added reference the map (above), we have added one new reference (Reference 28) in-text in the methodology and to the reference list while expanding the section on sample size (Page 6, lines 128-130, and Page 28, lines 653-655 respectively). We have also cross-checked to ensure all references are correctly cited.

TO REVIEWER #1 

(1) Is it "Screening and treatment practices for IDA during pregnancy among ANC visitors or among anaemic pregnant mothers?" Clearly state and write it from the result and discussion sections depending on your aim.

RESPONSE

Thank you for this observation. The title has been revised to enhance clarity and in accordance with the study objectives, the results and discussion. We have changed it from “Screening and treatment practices for iron-deficiency anaemia during pregnancy: a cross-sectional survey of healthcare workers in Nigeria” to “Screening and treatment practices for iron-deficiency in anaemic pregnant women: a cross-sectional survey of healthcare workers in Nigeria” (Page 1, lines 1-3).This has also been aligned with the objectives for enhance clarity (Page 2, line 31 under Abstract, and Page 5, line 99 under introduction).

(2)You state various keywords even more than the recommended.

RESPONSE

Thank you for drawing our attention to this. We were unsure of the maximum number of keywords allowed by the journal because we could not access the information, even now. However, we have now reduced the number of keywords from thirteen to eight (Page 3, lines 55-57). The keywords are listed in order of importance, so depending on the maximum allowed by the journal, the first five (for example) can be used if there is a need to reduce further.

(3) You used abbreviations/acronyms at the abstract section.

RESPONSE

Thank you for bringing forth this observation. We were trying to restrict word count to maximum 300 which was the information we had. We have now revised the abstract by writing acronyms/abbreviations in full (Abstract, Page 2, lines 26-46; and Page 3, lines 50-51). The abstract is now 314-words long. We hope this is fine. We also just came across another PLOS One guide which says 300 words preferred by journal but maximum 500 words acceptable for abstracts.

(4) Show clearly how you calculated sample size.

RESPONSE

Thank you. Details of sample size calculation has been added to the appropriate section under Methodology (Page 6, lines 124-141, and page 7, lines 142-152).

(5) Why do you consider an attrition rate of 30%? How much is the recommendation?

RESPONSE

Thank you for this question. Generally, in clinical research, 10-20% is often used to adjust for attrition during sample size calculation. To the best of our knowledge there is no benchmark. Considering that this is a survey of health workers, we decided to use a higher value to adjust for non-response to improve our sample size for the following reasons: (i) Surveys are generally prone to higher rate of attrition compared to other forms of clinical research, and (ii) a previous study found that doctors are less likely to respond to surveys compared to patients (53% vs. 70%, p <0.05) irrespective of the mode of administering the survey and that average response rate for in-person surveys is 76%.1 (iii) When attrition is above 20% in a study, the risk of bias increases.2 To minimize these potential issues, we decided to adjust the sample size calculation for non-response of 30%. 

Reference

1. Meyer VM, Benjamens S, Moumni ME, Lange JFM, Pol RA. Global Overview of Response Rates in Patient and Health Care Professional Surveys in Surgery: A Systematic Review. Annals of Surgery. 2022;275(1):e75-e81.

2. Dumville JC, Torgerson DJ, Hewitt CE. Reporting attrition in randomised controlled trials. Bmj. 2006;332(7547):969-71.

(6) Put your operational definitions.

RESPONSE

Thank you. This has been added to the methods section (Page 10, lines 234-236, and page 11, lines 237-251).

(7) More nurses/midwives (81.1%) compared to doctors (47.4%) reported lack of laboratory facilities for not screening for iron deficiency in anaemic pregnant women. Compared to doctors, a significantly higher percentage of nurses/midwives reported that they do not prescribe any treatment. Do you consider any knowledge and attitude issue in addition to referral practice?

RESPONSE

Thank you for this question. Yes, knowledge may be relevant in explaining this; generally, doctors have better exposure and thus knowledge regarding treatment of medical conditions. The curriculum for training doctors is more elaborate than that of the nurses. So, knowledge can partly explain why nurses are less likely to use the laboratory or prescribe treatment compared to doctor. We are not quite certain if attitude can play a role. Rather, health facility policies might contribute. Policies differ from facility to facility. Some health facilities may permit doctors to screen and treat certain conditions but not the nurses. Some of these comments not previously captured in the discussion have been added to the section (Page 23, lines 504-506).

(8) Why do you concern in low usage of intravenous iron than oral iron?

RESPONSE

Thank you. For this study, there is no specific concern. This is one of the sub-studies for my (lead author) doctoral research. The main study was a randomized controlled trial which evaluated the effectiveness of intravenous iron (ferric carboxymaltose) and oral iron (ferrous sulphate) for the treatment of iron deficiency anaemia during pregnancy. So, we were interested in knowing more about the usage rate of intravenous iron in Nigeria.

TO REVIEWER #2 

Thorough background research on the state of maternal health care workforce in Nigeria. Good job by the research team. More specific references for those values of the population of maternal healthcare professionals would be better.

RESPONSE

Thank you for this comment. Indeed, it would have been great if we were able to do so. Unfortunately, we tried to get data directly from the Medical and Dental Council of Nigeria for doctors, and Nurses and Midwifery Council of Nigeria for the nurses but did not succeed. So, we had to rely on available data. There is also no online source from which we could retrieve these data.

---

## [Decision Letter · Decision Letter 1]

9 Sep 2024

Screening and treatment practices for iron deficiency in anaemic pregnant women: a cross-sectional survey of healthcare workers in Nigeria

PONE-D-24-09943R1

Dear Dr. Babah,

We’re pleased to inform you that your manuscript has been judged scientifically suitable for publication and will be formally accepted for publication once it meets all outstanding technical requirements.

Within one week, you’ll receive an e-mail detailing the required amendments. When these have been addressed, you’ll receive a formal acceptance letter, and your manuscript will be scheduled for publication.

Kind regards,

Blessing Akombi-Inyang, Ph.D.

Academic Editor

PLOS ONE

Additional Editor Comments (optional):

Reviewers' comments:

Reviewer's Responses to Questions

**Comments to the Author**

1. If the authors have adequately addressed your comments raised in a previous round of review and you feel that this manuscript is now acceptable for publication, you may indicate that here to bypass the “Comments to the Author” section, enter your conflict of interest statement in the “Confidential to Editor” section, and submit your "Accept" recommendation.

Reviewer #1: All comments have been addressed

Reviewer #3: All comments have been addressed

2. Is the manuscript technically sound, and do the data support the conclusions?

Reviewer #1: Yes

Reviewer #3: Yes

3. Has the statistical analysis been performed appropriately and rigorously? 

Reviewer #1: Yes

Reviewer #3: Yes

4. Have the authors made all data underlying the findings in their manuscript fully available?

Reviewer #1: Yes

Reviewer #3: Yes

5. Is the manuscript presented in an intelligible fashion and written in standard English?

Reviewer #1: Yes

Reviewer #3: Yes

6. Review Comments to the Author

Reviewer #1: Thanks, all comments have addressed based on the comments i provided. so i have no concern if it will be accepted

Reviewer #3: (No Response)

7. PLOS authors have the option to publish the peer review history of their article (what does this mean?). If published, this will include your full peer review and any attached files.

Reviewer #1: **Yes: **Moges Tadesse Abebe

Reviewer #3: No

---

## [Editor Report · Acceptance letter]

19 Sep 2024

PONE-D-24-09943R1 

PLOS ONE

Dear Dr. Babah, 

I'm pleased to inform you that your manuscript has been deemed suitable for publication in PLOS ONE. Congratulations! Your manuscript is now being handed over to our production team.

Kind regards, 

on behalf of

Dr. Blessing Akombi-Inyang 

Academic Editor

PLOS ONE